# New Nitric Oxide-Releasing Compounds as Promising Anti-Bladder Cancer Drugs

**DOI:** 10.3390/biomedicines11010199

**Published:** 2023-01-12

**Authors:** María Varela, Miriam López, Mariana Ingold, Diego Alem, Valentina Perini, Karen Perelmuter, Mariela Bollati-Fogolín, Gloria V. López, Paola Hernández

**Affiliations:** 1Departamento de Genética, Instituto de Investigaciones Biológicas Clemente Estable, Avenida Italia 3318, Montevideo 11600, Uruguay; 2Laboratorio de Biología Vascular y Desarrollo de Fármacos, Institut Pasteur Montevideo, Mataojo 2020, Montevideo 11400, Uruguay; 3Cell Biology Unit, Institut Pasteur Montevideo, Mataojo 2020, Montevideo 11400, Uruguay; 4Departamento de Química Orgánica, Facultad de Química, Universidad de la República, Avenida General Flores 2124, Montevideo 11800, Uruguay

**Keywords:** bladder cancer, furoxans, nitric oxide donors, NF-κB, survivin

## Abstract

Bladder cancer is a worldwide problem and improved therapies are urgently needed. In the search for newer strong antitumor compounds, herein, we present the study of three nitric oxide-releasing compounds and evaluate them as possible therapies for this malignancy. Bladder cancer cell lines T24 and 253J were used to evaluate the antiproliferative, antimigratory, and genotoxic effects of compounds. Moreover, we determined the NF-κB pathway inhibition, and finally, the survivin downregulation exerted by our molecules. The results revealed that compounds **1** and **3** exerted a high antiproliferative activity against bladder cancer cells through DNA damage and survivin downregulation. In addition, compound **3** reduced bladder cancer cell migration. We found that nitric oxide donors are promising molecules for the development of a new therapeutic targeting the underlying mechanisms of tumorigenesis and progression of bladder cancer.

## 1. Introduction

Bladder cancer (BC) is one of the most common genitourinary malignancies and represents a serious health problem worldwide. Its incidence rises with age and it is three times more common in men than in women [1]. BC risk factors include genetic and molecular abnormalities, chemical or environmental exposures, and chronic irritation [2].

Cisplatin-based combination chemotherapy has been the first-line treatment for metastatic BC, providing an overall survival of 14–15 months and 5-year survival of 13–15% [3]. Yet, two-thirds of patients are ineligible due to impaired performance status or comorbidities [4] while 30% of patients do not respond to initial chemotherapy or have recurrence within the first year of treatment [5]. Over this past decade, the emergence of contemporary immunotherapy, targeted inhibitors, and antibody–drug conjugates has significantly changed the long-standing, predominantly chemotherapy-based option, giving a second wave of hope and prolonging overall survival [6]. Still, low objective response rates and poor survival demand further investigation of new and more effective therapeutic strategies for BC.

The selective inhibition of cell proliferation and the induction of apoptosis are crucial aspects of anticancer therapies. Nitric oxide (·NO)-releasing compounds alone or in combination with traditional chemo- or radiotherapy are promising agents for the treatment of BC [7]. ·NO can induce a multitude of antitumor effects such as the inhibition of cell proliferation; apoptosis stimulation; sensitization to chemo-, radio-, or immunotherapy; and the impairment of angiogenesis, invasion, and metastasis [8]. High local ·NO concentrations increase the intracellular content of nitrogen and oxygen-reactive species generating nitrosative and oxidative stress inducing cytotoxic effects in cancer cells [9]. Under this oxidative state, nitrous anhydride and peroxynitrite generated by ·NO are the major inductors of genotoxicity leading to the deamination of DNA bases, the oxidation of bases and deoxyribose, strand breaks, and multiple types of cross-linking events [10]. Moreover, ·NO can inhibit the nuclear factor kappa-B (NF-κB) by S-nitrosylation modulating its gene products [11]. Dysregulation of the NF-κB pathway plays an important role in cancer progression and metastasis by inducing gene transcription of growth-promoting, anti-apoptotic, and epithelial mesenchymal transition factors (EMT) [12,13]. In BC cells, it has been demonstrated that NF-κB activation enhances survivin expression which in turn promotes apoptosis resistance and cancer cell proliferation [14]. Elevated expression of survivin has been associated with an advanced cancer stage, poor prognosis, and decreased response to therapy [15,16]. In addition, survivin is an independent predictor of recurrence and cancer-specific survival [17]. Thus, therapies based on ·NO-releasing compounds represent an attractive approach for BC treatment.

In previous works, we showed the promising potential application of ·NO-releasing compounds for the treatment of BC [18]. Herein, continuing with our efforts to explore the anticancer action of arylsulfonylfuroxan derivatives, we performed the chemical synthesis of three new ·NO-releasing compounds (Figure 1) and evaluated their potential use as drug candidates in BC.

To assess these objectives, we employed two BC cell lines: T24 cells derived from a primary grade 3 transitional cell carcinoma [19] and 253J cells derived from a metastatic transitional cell tumor of the urinary tract [20]. We found that compounds **1** and **3** were excellent antiproliferative agents toward BC cells by inducing DNA damage and survivin downregulation. Both molecules showed an improved selectivity toward cancer cells compared to the drug cisplatin indicating the therapeutic potential of these compounds. The ·NO release exerted by these molecules was involved in the antiproliferative activity against cancer cells. However, compounds showed differences in their mode of action. Compound **1** showed an important inhibition of the NF-κB pathway and a moderate ability to downregulate the survivin level, while compound **3** behaved as a strong antiproliferative agent in both 2D and 3D cancer cell cultures, a cell migration inhibitor, and a potent survivin downregulator. Thus, compound **3** emerges as a promising molecule for BC treatment. Moreover, since the overexpression of survivin confers chemo- and radioresistance in a variety of human cancers [21,22], we highlight the use of ·NO-releasing compounds as a promising hit and scaffold for future drug design for cancer treatments in which survivin is upregulated. In this sense, this study opens the possibility for further approaches combining these ·NO donors with classical therapies, with the aim to sensitize tumor cells and enhance the efficacy of the drugs in clinical use.

## 2. Materials and Methods

### 2.1. General Experimental Information

Chemical supplies were from Sigma-Aldrich (Saint Louis, MO, USA). Compounds **1** [23], **4** [24], and **5** [24] were prepared as reported. Electron impact mass spectra (MS) were performed on a Shimadzu GC–MS QP 1100 EX instrument and high-resolution mass analysis was performed on a Thermo Scientific Q Exactive Hybrid Quadrupole-Orbitrap Mass Spectrometer using MeOH as a solvent. ^1^H NMR and ^13^C NMR spectra were obtained on a Bruker Avance DPX-400 spectrometer, using TMS as internal reference. The chemical shifts (δ) are reported in parts per million (ppm) relative to the center line of the corresponding solvent. The reaction progress was analyzed by TLC (silica gel 60F-254 plates visualized with UV light (254 nm)). Column chromatography was carried out using silica gel (230–400 mesh). 

Cell culture supplies were from Biological Industries (Beit Haemek, Israel) and Capricorn Scientific (Ebsdorfergrund, Germany). Sulforhodamine B, cisplatin, and propidium iodide were from Sigma-Aldrich (St. Louis, MO, USA). SAHA was from Acade (Hong Kong, China). Protease inhibitors were purchased from Roche. Antibodies against survivin (ab76424) and alpha-tubulin (ab15246) and the ECL chemiluminescence kit were from Abcam (Cambridge, MA, USA). The secondary antibody goat-anti-rabbit HRP (G21234) was from Invitrogen (Rockford, IL, USA) and the PVDF membrane (RPN303F) was from GE Healthcare Life Sciences (Little Chalfont, Buckinghamshire, UK). The bladder cancer cell line T24 (ATCC HBT-4) was purchased from the Cell Repository ABAC (Asociación Banco Argentino de Células). The bladder cancer cell line 253J and immortalized human keratinocyte cell line HaCaT (BCRJ batch number 001071) were kindly provided by Dr. Wilner Martínez-López and Dr. Jimena Hochmann, respectively.

### 2.2. Experimental Procedures and Characterization Data for the Compounds

3-(3-Phenylsulfonyl-*N*^2^-oxide-1,2,5-oxadiazole-4-oxy)propyl 6-((3-carboxypropanoyl)oxy)-2,5,7,8-tetramethylchroman-2-carboxylate (**2**). A solution of 3-(3-phenylsulfonyl-*N*^2^-oxide-1,2,5-oxadiazole-4-yl)oxypropyl 6-hydroxy-2,5,7,8-tetramethylchroman-2-carboxylate (4, 0.090 mmol), succinic anhydride (0.135 mmol), and cesium carbonate (0.180 mmol) in 1.5 mL acetonitrile was stirred at room temperature for 24 h and the completion of the reaction was monitored by TLC. The solvent was evaporated under reduced pressure and the reaction crude was diluted with a 10% HCl solution and extracted with EtOAc. The combined organic layers were dried with Na_2_SO_4_ and filtered, and the solvent was evaporated under reduced pressure. The residue was purified by flash column chromatography (SiO_2_, Hexane/EtOAc, 7/3) to render the desired product as a colorless oil that crystallized at 4 °C, yield 59%. ^1^H and ^13^C NMR spectra are provided in the Appendix A. ^1^H NMR (400 MHz, CDCl_3_) δ 8.04 (d, J = 7.4 Hz, 2H), 7.75 (t, J = 7.5 Hz, 1H), 7.61 (t, J = 7.9 Hz, 2H), 4.40–4.35 (m, 1H), 4.16–4.02 (m, 3H), 2.91 (bs, 2H), 2.82 (bs, 2H), 2.65–2.60 (m, 1H), 2.51–2.45 (m, 2H), 2.16 (s, 3H), 2.09–2.04 (m, 2H), 2.01 (s, 3H), 1.95–1.92 (m, 1H), 1.89 (s, 3H), 1.65 (s, 3H). ^13^C NMR (101 MHz, CDCl_3_) δ 173.9, 173.7, 171.5, 156.7, 149.5, 138.1, 135.6, 129.7, 128.5, 127.4, 125.3, 122.9, 117.1, 110.5, 77.5, 67.3, 60.4, 30.9, 30.5, 28.5, 20.9, 12.9, 12.0, 11.8. MS (IE, 70 eV) *m/z* (%): 532 (M^+^-Succ.Ac., 12), 516 (17), 391 (M^+^-Fx, 10), 333 (2), 232 (7), 217 (6), 205 (91), 142 (28), 77 (100). HRMS (ESI+): *m/z* calculated for C_29_H_32_N_2_O_12_SNa: 655.1574 [M + Na]^+^; found 655.1608.

6-((3-carboxypropanoyl)oxy)-2,5,7,8-tetramethyl-*N*-[2-(3-phenylsulfonyl-*N*^2^-oxide-1,2,5-oxadiazole-4-yl)oxyethyl]chroman-2-carboxamide (**3**). The title compound was prepared from 6-hydroxy-2,5,7,8-tetramethyl-*N*-[2-(3-phenylsulfonyl-*N*^2^-oxide-1,2,5-oxadiazole-4-yl)oxyethyl]chroman-2-carboxamide (**5**, 0.200 mmol), succinic anhydride (0.400 mmol), and cesium carbonate (0.3 mmol) in 3 mL of DMF with stirring at room temperature for 3 h; the completion of the reaction was monitored by TLC. Subsequently, the reaction mixture was diluted with a 10% HCl solution and extracted with ethyl ether. The combined organic layers were dried with Na_2_SO_4_, filtered, and the solvent evaporated under reduced pressure. The residue was purified by flash column chromatography (SiO_2_, Hexane/EtOAc, 4/6) to render the desired product as a white solid, m.p. 79-81 °C, yield 30%. ^1^H and ^13^C NMR spectra are provided in the Appendix A. ^1^H NMR (400 MHz, acetone-d_6_) δ 8.10 (d, J = 7.4 Hz, 2H), 7.88–7.84 (m, 1H), 7.72–7.69 (m, 2H), 4.54 (bs, 2H), 3.80–3.70 (m, 2H), 2.97–2.93 (m, 2H), 2.77–2.74 (m, 2H), 2.66–2.54 (m, 2H), 2.36–2.28 (m, 1H), 2.17 (s, 3H), 2.02 (s, 3H), 1.98 (s, 3H), 1.92-–1.85 (m, 1H), 1.52 (s, 3H). ^13^C NMR (101 MHz, acetone-d_6_) δ 174.0, 172.6, 170.7, 159.3, 148.1, 141.7, 138.1, 135.8, 129.8, 128.6, 127.3, 125.6, 122.5, 118.1, 110.7, 78.5, 70.0, 38.6, 28.4, 28.2, 23.1, 20.0, 12.2, 11.3. MS (IE, 70eV) *m/z* (%): 375 (M^+^-Fx, 4), 275 (16), 203 (2), 141 (27), 101 (2), 77 (100). HRMS (ESI+): *m/z* calculated for C_28_H_31_N_3_O_11_SNa: 640.1577 [M + Na]^+^; found 640.1592.

### 2.3. Cell Cultures

Human bladder cancer cells T24 (derived from transitional cell carcinoma) and 253J (developed from a retroperitoneal metastasis) were grown in McCoy’s 5A medium (Biological Industries) supplemented with 10% FBS (Capricorn Scientific). The non-cancer keratinocyte cell line HaCaT was grown in DMEM (Biological Industries) and the reporter cell line HT-29-NF-κB-hrGFP was grown in RPMI supplemented with 10% FBS (Capricorn Scientific). The cells were passaged twice per week, and the culture medium was changed with the same frequency. Cell cultures were maintained under humidified 5% CO_2_ atmosphere at 37 °C. 

### 2.4. Antiproliferative Activity

The antiproliferative effect of compounds in BC cells and in the non-cancer cells HaCaT was evaluated as follows. T24 and 253J cells (8 × 10^3^ cells per well) and HaCaT keratinocytes (10 × 10^3^ cells per well) were seeded in a 96-well plate and allowed to attach for 24 h. Afterward, the culture media was removed and the solubilized compounds in DMSO were added at increasing concentrations (0.1–50 µM) diluted in fresh culture medium in triplicate. The reference anticancer compounds cisplatin and suberoylanilide hydroxamic acid (SAHA) were included in each experiment. The cells were further incubated at 37 °C and 5% CO_2_ for 24 h. Then, the antiproliferative activity of the compounds was determined using the sulforhodamine B method [25]. Absorbance was measured at 510 nm and background at 620 nm using a microplate spectrophotometer (Varioskan Flash Microplate spectrophotometer, Thermo Fisher, Waltham, MA, USA). The IC_50_ was determined as the concentration that reduces absorbance by 50% compared with the control 0.5% DMSO and was determined by linear regression analysis. Each assay was repeated at least three times.

### 2.5. NO Release in Cell Culture Media

T24 and 253J cells (8 × 10^3^ cells per well) were seeded in a 96-well plate and allowed to attach for 24 h. Culture media were removed and the solubilized compounds in DMSO were added at 50 µM diluted in fresh culture medium in sextuplicate. The control with 0.5% DMSO was included in the experiments. After that, cells were incubated at 37 °C and 5% CO_2_ for 3 h. Then, the ·NO production as the nitrate/nitrite content was measured by the Griess reaction assay [26]. Briefly, 50 μL of culture medium was transferred to a new 96-well plate. Subsequently, 50 μL of 1% sulfanilamide solution was added to each well and incubated for 10 min protected from light. Finally, 50 μL of 0.1% N-1-naphtilethylenrdiamine dihydrochloride solution was added and incubated for an additional 10 min. A reference curve with NaNO_2_ was performed at serial dilutions between 0 and 100 μM in 50 μL of culture medium. Absorbance was measured at 540 nm using a microplate spectrophotometer.

### 2.6. NO Release in Physiological Solution

NO released by the compounds during incubation in a physiological solution in the presence of L-cysteine was determined [27]. The compounds were solubilized in DMSO and then diluted at 50 μM in a mixture of 50 mM pH 7.4 PBS solution/MeOH (1% DMSO) 50/50 *v*/*v*. Afterward, they were incubated in the presence of L-cysteine at a 0.25 mM concentration (a 5-fold excess compared to the ·NO-donor derivative) for 1, 3, and 6 h at 37 °C. The presence of nitrite in the sample was determined using the Griess reaction assay. At the same time, a standard curve with NaNO_2_ was performed at serial dilutions between 0 and 100 µM in 50 µL of the PBS/MeOH mixture. The absorbance was measured at 540 nm using a microplate spectrophotometer.

### 2.7. Antiproliferative Activity with Hemoglobin

To investigate the contribution of antiproliferative–proliferative activity of the studied compounds, we evaluated their effect on T24 and 253J bladder cancer cell culture’s growth in the absence and presence of Hb. Cells were seeded into a 96-well plate at 8 × 10^3^ cells per well and were allowed to attach for 24 h. The cultures were pretreated with hemoglobin (Hb) at 0 µM or 50 µM for 1 h and then treated with 50 µM of the selected compounds for 24 h. Then, the antiproliferative activity was assessed by the sulforhodamine B assay [18]. Statistical analysis was carried out using a two-way ANOVA analysis test followed by Bonferroni’s multiple comparison test.

### 2.8. Clonogenic Assay

Compounds were tested for their ability to inhibit T24 and 253J bladder cancer cell colonies. Cells were seeded (500 cells per well) in 60 mm plates and allowed to attach for 5 h after seeding. Then, compounds were added at a final concentration of 10 μM. The control with 0.5% DMSO was included in the experiments After 6 days, the clones were fixed with a solution of acetic acid in methanol 1%, stained with crystal violet 0.5%, and counted. The colonies formed (CF) were counted and then the plating efficiency (PE) and the survival fraction (SF) were calculated for each compound. The PE was calculated by dividing the CF by the number of cells plated. The SF was determined by dividing the PE of the treated cells by the PE of the control cells and then multiplying by 100 [28]. 

### 2.9. Spheroids 

To evaluate the effects of our compounds in 3D spheroids, 96-well plates were pretreated with 1.5% agarose in PBS. Then, cells were seeded (3 × 10^4^ cells per well) in 200 µL of cell culture media. The culture was placed in culture conditions for 3 days to allow spheroid production. At day 3, 100 µL of the media was removed carefully and replaced with 100 µL 2X of the final concentration of our compounds. DMSO was used as a control. Finally, at 24, 48, and 72 h, the culture media were replaced by resazurin 2× in PBS and fluorescence at 530 nm (ex) and 590 nm (em) was measured in a microplate reader [29].

### 2.10. Scratch Assay

The ability to inhibit the cancer cell migration was evaluated using a scratch assay. T24 and 253J cells were seeded at 5 × 10^5^ cells per well in six-well plates and incubated for 24 h. Cell culture media were removed and washed with PBS. Cultures were incubated with mitomycin-c at 5 µg/mL in a culture medium supplemented with 1% SFB for 2 h. Then, cells were washed with PBS and wounds were created by scratching cell monolayers with a sterile 200 μL plastic pipette tip. Compounds were added at 5 and 10 µM in fresh culture media supplemented with 1% SFB for 17 h. Six images per well were taken in a phase contrast microscopy Olympus IX-81 with a 10× objective at 0 and 17 h and quantified using the ImageJ program [30]. Statistical analysis was performed using a one-way ANOVA test followed by Dunnett’s test.

### 2.11. Comet Assay

The genotoxic damage induced in T24 and 253J cells after 3 h of treatment with compounds was evaluated by alkaline single-cell gel electrophoresis (comet assay). Cells were seeded in 35 mm plates at 2 × 10^5^ cells per well and allowed to attach for 24 h in a humidified 5% CO_2_ atmosphere at 37 °C. Afterward, the culture media were removed and compounds were added at 50 µM in a fresh culture medium. Hydrogen peroxide at 100 µM was included a positive control. The cells were further incubated for 3 h at 37 °C and 5% CO_2_. After the incubation step, cells were detached and centrifuged for 10 min at 1200 rpm and the cell pellet was resuspended in PBS. Cell suspensions were mixed with 1.5% low-melting-point agarose and immediately placed on slides pre-coated with normal-melting-point agarose. The agarose was allowed to set for 15 min at 4 °C and cells were lysed through the immersion of slides in a cold solution of lysis buffer (2.5 M NaCl, 100 mM Na_2_EDTA, 10 mM Trizma-HCl, NaOH to pH 10 and 1% Triton X-100) for 12 h at 4 °C. Slides were incubated in a cold alkaline electrophoresis solution (300 mM NaOH and 1 mM Na_2_EDTA, pH 13) for 20 min to DNA unwinding and expression of alkali labile sites were allowed. Electrophoresis was performed at 25 V (300 mA) for 20 min in a cold unit at 4 °C. Samples were washed three times with neutralization buffer pH 7.5 (0.4 M Tris–HCl). Finally, slides were stained with bromodeoxyuridine (10 µg/mL) [31]. Images were taken using an epifluorescence microscopy Olympus IX-81 with a 20× objective. A total of 100 comets on each comet slide were visually scored and classified as belonging to one of the five classes according to the tail length and given a value from 0 (undamaged) to 4 (maximum damage). The DNA damage score in the range of 0 to 400 was calculated using the equation: Σ(n.α), where n is the percentage of cells in a class of tail length and α is the class of tail length [32,33].

### 2.12. NF-κB Pathway Inhibition

To determine if these ·NO-releasing compounds inhibit the NF-κB pathway, we used the reporter cell line HT-29-NF-κB-hrGFP as we previously described [34]. 

A—HT-29-NF-κB-hrGFP cells cytotoxicity assay: 4 × 10^4^ cells per well were seeded in a 96-well plate in RPMI and incubated for 24 h at 37 °C and 5% CO_2_. The culture media were removed and the compounds, dissolved in DMSO (lower than 0.5%, v/v, in the final volume of RPMI), were added at the desired final concentrations diluted in fresh RPMI (0.19–50 µM) and the cells were further incubated for 24 h at 37 C, 5% CO_2_. Afterward, the culture supernatant was removed and cell viability was determined by the MTT assay. Absorbance measurements were performed at 570 nm in a spectrophotometer plate reader.

B—NF-κB pathway inhibition assay: Cells were seeded in a 96-well plate (4 × 10^4^ cells per well) in a fresh RPMI culture media and incubated for 24 h. Cell cultures were pre-treated with compounds at 5 and 10 µM for 1 h prior to pro-inflammatory stimuli with TNF-α (1 ng/mL) and cultures were further incubated for 24 h. Three controls were included: cells treated only with TNF-α, cells treated only with compounds, and cells treated with BAY 11-7082 at 10 µM and TNF-α (1 ng/mL). Cells were detached and resuspended to perform flow cytometry analysis. Cells were analyzed using a BD Accuri™ C6 (Biosciences, BD, USA) flow cytometer equipped with 488 and 640 nm lasers. Data acquisition and analysis was achieved using BD Accuri C6 Software V1.0.264.21. The GFP and propidium iodide fluorescence emissions were detected using band-pass filters 533/30 and 585/40, respectively. For each sample, 5000 counts gated on an FSC versus SSC dot plot (excluding doublets) were recorded. For analysis, only single living cells (those that excluded propidium iodide) were considered. The percentage of GFP positive was normalized against the percentage of GFP cells obtained with the TNF-α control. Statistical analysis was carried out using a one-way ANOVA test and Dunnett’s test.

### 2.13. Western Blotting

The capability of compounds to reduce survivin levels in BC cells was determined through a Western blot. T24 or 253J cells (5 × 10^5^ cells) were seeded in 10 cm Petri dishes in a complete medium and allowed to attach for 24 h. Cells were treated with compounds **1** and **3** at 10 µM or 0.5% DMSO for 24 h. At the end of the incubation period, cells were washed twice with PBS 1X and harvested. Proteins were solubilized in Laemmli buffer plus protease inhibitors (NaF, PMSF and protease inhibitor mixture) followed by denaturation at 95 °C for 5 min and resolved on a 15% SDS-PAGE. For immunoblotting, proteins were electro-transferred onto a PVDF membrane and blocked with 5% nonfat milk in TBS-0.10% Tween 20 (TBST) at room temperature for 1 h. Then, the membrane was incubated with the primary antibodies rabbit anti-survivin (1/5000) and rabbit tubulin (1/200) at 4 °C overnight. After washing (TBST) and subsequent blocking, the blot was incubated with the secondary antibody goat-anti-rabbit HRP (1/5000) for 1 h at room temperature. The blots were developed using an enhanced chemiluminescence ECL detection reagent. Quantification was performed with ImageJ software. Statistical analysis was carried out using two-way ANOVA followed by a Bonferroni multiple comparison test. 

## 3. Results

### 3.1. Chemistry

#### Chemical Synthesis

Furoxan **1** was synthesized as previously [23] and furoxan derivatives **2** and **3** were synthesized from **4** and **5** [24] by treatment with succinic anhydride in the presence of cesium carbonate as illustrated in Figure 1. 

### 3.2. Biology

#### 3.2.1. Antiproliferative Activity

The in vitro antiproliferative activity of the compounds was determined in bladder-cancer-derived cell lines T24 and 253J and in the non-cancer HaCaT cells to determine the cytotoxic selectivity index against cancer cells. The cells were incubated in the presence of the compounds for 24 h. The antiproliferative activity was determined using the sulforhodamine B assay. The results presented in Table 1 are expressed as IC_50_ in µM, which is the drug concentration resulting in a 50% reduction in cellular net growth when compared with the negative control DMSO 0.5%. The standard anticancer drug cisplatin and SAHA were used as positive controls.

Compounds **1**, **2,** and **3** showed an important antiproliferative activity in T24 and 253J cancer cell lines. As an attempt to approximate the safety profile and selectivity toward cancer cells, we further studied the antiproliferative activity in non-cancerous HaCaT cells. The selectivity index (SI) was calculated as the ratio between the IC_50_ for HaCaT cells and the IC_50_ for BC cells. The results show that compounds **1** and **3** provided a higher selectivity for cancer cells than the chemotherapeutic drug cisplatin. It is worth mentioning that compound **3** exhibited the highest antiproliferative effect on the cisplatin-resistant cell line 253J and a 20-fold higher SI value 20 than cisplatin. Thus, this study suggests that compounds **1** and **3** presented the best profile for further studies in BC cells.

#### 3.2.2. Nitric-Oxide-Releasing Activity

Furoxan heterocycles are antitumor pharmacophores that are capable of releasing ·NO via a thiol-dependent mechanism [35]. In this regard, the ·NO-releasing activity of compounds was determined at 50 µM incubated for 3 h in T24 and 253J cell cultures was determined by the Griess assay. The strong nitric-oxide-releasing compound SNAP was used as a control.

The results indicate that the compounds were able to release nitric oxide in cell culture media after 3 h of incubation (Figure 2). Compound **3** produced higher ·NO levels in T24 cell culture, while compound **2** produced higher ·NO levels in 253J cell culture. Compound **1** induced the lowest nitrate/nitrite levels.

The ·NO production by furoxans at 50 µM was also measured in physiological solution in the presence of L-cysteine (5-fold molar excess) after incubation for 1, 3, and 6 h at 37 °C. 

The results indicated that in the physiological solution, the compounds **1**, **2**, and **3** and SNAP released higher ·NO levels at 3 h as we indicated previously [18] (Figure 3). Moreover, the results showed that the ·NO-releasing capacity of the products in the cell milieu differed from that under physiological conditions, suggesting that they exhibited different physicochemical properties and reactivity towards thiols.

#### 3.2.3. Antiproliferative Activity with Hemoglobin

The contribution of ·NO to the antiproliferative activity of the studied compounds was evaluated on BC cells grown in the absence and presence of hemoglobin. Cultures were pretreated with or without hemoglobin (50 µM) for 1 h and then treated with compounds (10 µM) for 24 h until the sulforhodamine B assay. The results indicated that the antiproliferative activity of the compounds decreased in the presence of an ·NO scavenger (hemoglobin), suggesting that the ·NO release was involved in their anticancer mechanism (Figure 4).

We observed that the antiproliferative activity of the compounds **1**, **2,** and **3** in T24 cells was mainly due to ·NO-releasing activity. On the other hand, in 253J cells, the ·NO-releasing activity was partially involved in the antiproliferative effect. Therefore, these results suggest that the invasive BC cell line T24 is highly sensitive to ·NO while the cisplatin-resistant cell line 253J is moderately sensitive to ·NO.

#### 3.2.4. Clonogenic Assay

A clonogenic assay was performed in T24 and 253J cells. The results are shown in Figure 5 and the SFs determined by the ratio between the PE of the treated and control cells are presented in Table 2. Compounds **1** and **3** were able to inhibit the clonogenic ability of both cancer cell lines. Compound **2** only inhibited the clonogenic survival in 253J cells. According to these results and the antiproliferative activity, we selected compounds **1** and **3** for further studies.

#### 3.2.5. Spheroids

The capacity of the compounds to inhibit cell growth in 3D cultures was determined in T24 cancer cell spheroids. In this assay, spheroids of T24 cells were incubated with compounds **1** or **3** or cisplatin for 24, 48, and 72 h. Cell viability was then determined using the resazurin method. The results indicate that compound **3** had a strong inhibitor effect on 3D cell growth (Figure 6). Even at 24, 48, and 72 h, this compound exerted a higher antiproliferative activity than the reference compound cisplatin. 

The fact that the potent antiproliferative activity of compound **3** increased over the time suggests that BC cells are unable to avoid its anticancer effect. 

#### 3.2.6. Migration Assay

The ability to inhibit cell migration was determined using a scratch assay. Considering the above results, **1** and **3** were selected to perform this study. To suppress cell proliferation, cells were incubated with mitomycin-c for 2 h and then removed by washing before making the scratch. Images were immediately taken (0 h). Cells were incubated for 17 h in the presence of compounds **1** and **3** at 5 and 10 µM. SAHA at 10 µM, a pan-HDAC inhibitor, was included as a control. After this period, images were captured to measure gaps. results indicate (Figure 7) that in T24 cells, compound **3** was the most potent compound able to reduce cell migration (57%), even better than SAHA (47%). On the other hand, in 253J cells, SAHA reduced cell migration by 60%, whereas compound **3** reduced cell migration by 30%.

#### 3.2.7. Comet Assay

The induction of genotoxic effect by ·NO-releasing compounds was assessed by measuring DNA migration caused by strand breaks in the alkaline single-cell electrophoresis assay. BC cells were incubated in the presence of compounds **1** and **3** at 50 µM or H_2_O_2_ at 100 µM for 3 h. The results indicate that ·NO-releasing molecules induced DNA strand breaks in cancer cells (Figure 8). Compound **1** induced mainly class 3 DNA damage in both cell lines. Compound **3** also induced strand breaks, although the effect in T24 cells was lower than 253J cells. While in T24 this compound induced different degrees of DNA damage, in 253J cells, we identified comets that corresponded mainly to class 3. The positive control H_2_O_2_ induced a higher genotoxic effect with a high prevalence of class 4 DNA damage score.

#### 3.2.8. NF-κB Pathway Inhibition

Using a human pathway-specific reporter cell system (HT-29-NF-κB-hrGFP), the in vitro capability to decrease the activation of NF-κB level was determined. Human intestinal cells HT-29 stably transfected with the pNF-κB-hrGFP plasmid were stimulated with the pro-inflammatory cytokine TNF-α. NF-κB activation was estimated by measuring the percentage of GFP-expressing cells. Determinations were performed in the absence and presence of compounds **1** and **3** at the non-cytotoxic doses of 5 and 10 µM (100% of cell survival on HT-29-NF-κB-hrGFP cells). BAY-117082 at 10 µM was used as a reference compound. Controls without TNF-α stimulation were also included to evaluate the compounds’ intrinsic pro-inflammatory properties, finding that compounds were not pro-inflammatory per se.

The results shown in Figure 9 demonstrated that compound **1** at 10 µM inhibited NF-κB activation almost as well as the positive control BAY-117082. On the other hand, compound **3** was found to be inactive at the evaluated doses. Thus, compound **1** may suppress cancer cell proliferation by modulating the NF-κB pathway.

#### 3.2.9. Survivin Expression Inhibition

Survivin is known to be involved in cancer cell proliferation and apoptosis inhibition. Thus, inhibition of survivin expression holds promise for BC therapies. T24 and 253J cells were exposed to compounds **1** and **3** at 10 µM for 24 h and a Western blot assay was performed. Our results suggest that compound **1** is a moderate inhibitor of survivin expression in both cell lines (Figure 10). Taking into consideration the above results, compound **1** could induce the antiproliferative activity by modulating the NF-κB and reducing the survivin expression. On the other hand, compound **3** showed a strong downregulation of survivin level in both BC cell lines. Therefore, compound **3** emerges as a promising survivin inhibitor for cancer treatment.

## 4. Discussion

BC is one of the most frequent cancers and a worldwide health problem. There is an urgent need to develop new therapeutic agents targeting the underlying mechanisms that trigger the tumorigenesis and progression of BC. The main antitumor effects of ·NO include apoptosis, necrosis, and cytotoxicity stimulation; the inhibition of tumor cell proliferation; DNA fragmentation; the inhibition of NF-κB and the modulation of its gene products; and the impairment of angiogenesis and metastasis [8]. Continuing our efforts in the search for new arylsulfonylfuroxans with potent anticancer activity, herein, we describe the study of three ·NO-donor compounds as potential therapeutic agents for BC. Our results indicate that compounds **1**, **2,** and **3** exert a high antiproliferative activity against BC cells. 

In previous studies, in the BC cell lines described here, compounds analogs to **2** and **3**, including **4** and **5**, and the furoxan precursor of the latter were tested [18]. From these studies, compound **4** was found to have the highest antiproliferative activity and selectivity towards tumor cells in the 253J cell line (Table 3), while for the T24 cell line, compound **5** stands out, with an acceptable IC_50_ but with a relatively low selectivity index (Table 3). In the search to improve these results and to further study the underlying mechanisms of action, we started to work on the synthesis of hydroxamic acid derivatives of these compounds. However, the synthesis was very challenging and it was not possible to obtain these derivatives, but it was possible to obtain the synthetic intermediates described here, **2** and **3**, which we studied as shown, and achieved comparable-to-better activities than their precursors **4** and **5**. 

In this line of thought, if we compare the results of antiproliferative activity in the 253J cell line for compounds **1**, **4**, and **2**, going from a simple to a more complex arylsulfonylfuroxan, incorporating a tocopherol-mimetic pharmacophore with known antitumor activity [36,37], it was observed that compound **1** stood out for its antiproliferative activity and selectivity and was therefore selected for further studies. The proposed structural changes did not significantly improve its activity or its selectivity. In contrast, when we performed the same analysis on the T24 cell line for compound **3** and its precursors, it was observed that the structural changes introduced improved both its antiproliferative activity and selectivity (Table 3). Moreover, compound **3** also stood out for its antiproliferative activity in the 253J cell line, and was therefore selected for further studies.

Compounds **1** and **3** proved to have a higher selectivity for T24 and 253J cancer cells than for non-cancer cells HaCaT, surpassing the chemotherapeutic drug cisplatin. Compound **2** showed a high inhibition of cell growth in 253J cells and a moderate effect in T24 cells; however, its selectivity toward BC cells was low, indicating possible side effects. In addition, we observed that the ·NO-releasing activity of compounds was primarily responsible for the anticancer effect in T24 cells, whereas in 253J cells, the ·NO-releasing capacity was partially involved. These observations confirm previously reported studies that emphasise the importance of ·NO donors for BC [7,18]. A clonogenic assay showed similar anticancer behavior of compounds as was observed in the sulforhodamine B assay, indicating that compounds **1** and **3** are suitable for further studies in BC cells. In this sense, we performed a three-dimensional antiproliferative assay as an approach to determine the antitumor effect of compounds in a small tumor. As we have shown above (Figure 6), compound **3** exhibited higher toxicity towards cancer cells, increasing its antiproliferative effect between 24 and 72 h of treatment. Thus, compound **3** appears to be a promising molecule to perform future in vivo studies for BC. With the aim to identify possible mechanisms of action exerted by compounds **1** and **3**, additional studies were performed. It is well-known that ·NO can react with molecular oxygen to generate secondary species which induce DNA damage. Via an alkaline comet assay, we observed the induction of DNA strand breaks in the BC nucleus after incubation with **1** and **3** (Figure 8). However, this assay did not discriminate between alkali-labile sites and single- or double-strand breaks. Further studies will be performed to elucidate precisely the DNA damage induced by these ·NO donors. As we pointed out before, the NF-κB pathway has been described as an EMT and metastasis positive regulator [13]. Moreover, NF-κB activation upregulates survivin expression enhancing the proliferation and resistance to the apoptosis of BC cells [14]. In this regard, inhibition of the NF-κB pathway and downregulation of the growth-promoting and anti-apoptotic protein survivin is suitable for cancer therapy. Our results suggest that compound **1** inhibited the NF-κB pathway, slightly reduced cell migration, and moderately reduced survivin expression in BC cell lines. On the other hand, compound **3** significantly decreased cell migration and strongly downregulated survivin in BC cells, although it did not seem to affect the NF-κB pathway. Therefore, this molecule could impair the initial stage of the EMT process and reduce the survivin intracellular level through a different mechanism than compound **1**. Survivin plays an important role in DNA double-strand-break repair regulation through its interaction with proteins involved in double-strand-break repair machinery [38]. In this way, our results indicate that the synthetized ·NO-releasing compounds **1** and **3** induce DNA strand breaks and avoid DNA repair through survivin downregulation. Hence, in this work, we evidenced that the ·NO donors **1** and **3** inhibit BC cell proliferation with high selectivity and induce DNA damage through a mechanism that involves ·NO-releasing capacity and survivin downregulation. A general scheme of the identified mechanisms of action exerted by ·NO donors **1** and **3** in this work can be seen below (Figure 11). We highlight compound **3** as a promising therapy for BC. In the future, further studies of this ·NO donor will be performed.

Finally, herein, we confirmed that the generation of arylsulfonylfuroxans and their hybridization with tocopherol-mimetic pharmacophores is a promising strategy for BC treatment. The combined administration of these ·NO-releasing compounds and traditional cancer therapies must be considered to enhance its efficacy. To the best of our knowledge, this is the first report of ·NO-releasing compounds with survivin-downregulating activity proposed for BC treatment. Future approaches of ·NO donors in other cancer types in which survivin is upregulated will be performed. 

## 5. Conclusions

In conclusion, we report three ·NO-releasing compounds as promising anti-bladder-cancer drugs. We demonstrate that compound **3** is a potent inhibitor of cell proliferation and a strong downregulator of survivin levels in BC cells. Therefore, we believe that the development of ·NO donors is a promising therapeutic directed against the underlying mechanism of BC development.

## Data Availability

Not applicable.

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
