# Peer review of "New Nitric Oxide-Releasing Compounds as Promising Anti-Bladder Cancer Drugs"

_biomedicines, 2023, doi:10.3390/biomedicines11010199_

Round 1
Reviewer 1 Report
The manuscript was prepared very well. The introduction section justifies the purpose of the study. I congratulate the authors for the preparation of the manuscript
I would like to congratulate the authors for the structure of the manuscript and all the research carried out. It is highly publishable. However, there are some concerns, in part important, so the review articles need revision, see below.
Introduction
Why is this study considered relevant?
Are there any other types of disease that the study treatment is used for?
Would it be compatible with other therapies for prostate cancer?
Materials and Methods
do not require any input
Have you considered any testing for adverse effects?
Results
The tables/figures and the text describing them do not require any input, it is the strongest part of this study. although it could more clearly describe the results
Are the figures your own creation?
Discussion
- Include a section on strengths / limitations.
- It is possible to describe more mechanisms responsible for the described actions of
nitric oxide-releasing compounds
- What does this article contribute to, the authors should make their own assessment and include their own discussion of the results shown in the manuscript?
Conclusion
In the Conclusion section, state the most important outcome of your work. Do not simply summarize the points already made in the body — instead, interpret your findings at a higher level of abstraction. Show whether, or to what extent, you have succeeded in addressing the need stated in the Introduction (or objectives).
Reviewer 2 Report
In this paper entitled “New nitric oxide-releasing compounds as promising anti-bladder cancer drugs” aims to study three nitric oxide-releasing compounds and evaluated them as possible therapies for this malignancy. The use of Spheroids assay is a good principle. The paper is well written and has the makings of a publication. But, have the authors mention how is implemented? Side effects? A graphical resume should be provided as resume of main molecular pathway.
